# Nonlinear Acceleration of Stochastic Algorithms

**Damien Scieur**
INRIA, ENS,
PSL Research University,
Paris France
damien.scieur@inria.fr

**Francis Bach**
INRIA, ENS,
PSL Research University,
Paris France
francis.bach@inria.fr

**Alexandre d'Aspremont**
CNRS, ENS,
PSL Research University,
Paris France
aspremon@ens.fr

## Abstract

Extrapolation methods use the last few iterates of an optimization algorithm to produce a better estimate of the optimum. They were shown to achieve optimal convergence rates in a deterministic setting using simple gradient iterates. Here, we study extrapolation methods in a stochastic setting, where the iterates are produced by either a simple or an accelerated stochastic gradient algorithm. We first derive convergence bounds for arbitrary, potentially biased perturbations, then produce asymptotic bounds using the ratio between the variance of the noise and the accuracy of the current point. Finally, we apply this acceleration technique to stochastic algorithms such as SGD, SAGA, SVRG and Katyusha in different settings, and show significant performance gains.

## 1 Introduction

We focus on the problem

$$\min_{x \in \mathbb{R}^d} f(x) \tag{1}$$

where $f$ is a $L$-smooth and $\mu$-strongly convex function with respect to the Euclidean norm, i.e.,

$$\frac{\mu}{2}\|y - x\|^2 \ \leq \ f(y) - f(x) - \nabla f(x)^T(y - x) \ \leq \ \frac{L}{2}\|y - x\|^2.$$

We consider a stochastic first-order oracle, which gives a noisy estimate of the gradient of $f(x)$, with

$$\nabla_\varepsilon f(x) = \nabla f(x) + \varepsilon, \tag{2}$$

where $\varepsilon$ is a noise term with bounded variance. This is the case for example when $f$ is a sum of strongly convex functions, and we only have access to the gradient of one randomly selected function. Stochastic optimization (2) is typically challenging as classical algorithms are not convergent (for example, gradient descent or Nesterov's accelerated gradient). Even the averaged version of stochastic gradient descent with constant step size does not converge to the solution of (1), but to another point whose proximity to the real minimizer depends of the step size [Nedić and Bertsekas, 2001; Moulines and Bach, 2011].

When $f$ is a finite sum of $N$ functions, then algorithms such as SAG [Schmidt et al., 2013], SAGA [Defazio et al., 2014], SDCA [Shalev-Shwartz and Zhang, 2013] and SVRG [Johnson and Zhang, 2013] accelerate convergence using a variance reduction technique akin to control variate in Monte-Carlo methods. Their rate of convergence depends on $1 - \mu/L$ and thus does not exhibit an accelerated rate on par with the deterministic setting (in $1 - \sqrt{\mu/L}$). Recently a generic acceleration algorithm called Catalyst [Lin et al., 2015], based on the proximal point method improved this rate of convergence, but the practical performances highly depends on the input parameters. On the other hand, recent papers, for example [Shalev-Shwartz and Zhang, 2014] (Accelerated SDCA) and

[Allen-Zhu, 2016] (Katyusha), propose algorithms with accelerated convergence rates, if the strong convexity parameter is given.

When $f$ is a quadratic function then averaged SGD converges, but the rate of decay of initial conditions is very slow. Recently, some results have focused on accelerated versions of SGD for quadratic optimization, showing that with a two step recursion it is possible to enjoy both the optimal rate for the bias and variance terms [Flammarion and Bach, 2015], given an estimate of the ratio between the distance to the solution and the variance of $\varepsilon$.

A novel generic acceleration technique was recently proposed by Scieur et al. [2016] in the deterministic setting. This uses iterates from a slow algorithm to extrapolate estimates of the solution with asymptotically optimal convergence rate. Moreover, this rate is reached *without prior knowledge of the strong convexity constant*, whose online estimation is still a challenge (even in the deterministic case [Fercoq and Qu, 2016]) but required if one wants to obtain optimal rates of convergence.

Convergence bounds are derived by Scieur et al. [2016], tracking the difference between the deterministic first-order oracle of (1) and iterates from a linearized model. The main contribution of this paper is to extend the analysis to arbitrary perturbations, including stochastic ones, and to present numerical results when this acceleration method is used to speed up stochastic optimization algorithms.

In Section 2 we recall the extrapolation algorithm, and quickly summarize its main convergence bounds in Section 3. In Section 4, we consider a stochastic oracle and analyze its asymptotic convergence in Section 5. Finally, in Section 6 we describe numerical experiments which confirm the theoretical bounds and show the practical efficiency of this acceleration.

## 2  Regularized Nonlinear Acceleration

Consider the optimization problem
$$\min_{x \in \mathbb{R}^d} f(x)$$
where $f$ is a $L-$smooth and $\mu-$strongly convex function [Nesterov, 2013]. Applying the fixed-step gradient method to this problem yields the following iterates
$$\tilde{x}_{t+1} = \tilde{x}_t - \frac{1}{L}\nabla f(\tilde{x}_t). \tag{3}$$

Let $x^*$ be the unique optimal point, this algorithm is proved to converge with
$$\|\tilde{x}_t - x^*\| \leq (1-\kappa)^t \|\tilde{x}_0 - x^*\| \tag{4}$$
where $\|\cdot\|$ stands for the $\ell_2$ norm and $\kappa = \mu/L \in [0,1[$ is the (inverse of the) condition number of $f$ [Nesterov, 2013]. Using a two-step recurrence, the *accelerated gradient descent* by Nesterov [2013] achieves the improved convergence rate
$$\|\tilde{x}_t - x^*\| \leq O\Big((1-\sqrt{\kappa})^t \|\tilde{x}_0 - x^*\|\Big). \tag{5}$$

Indeed, (5) converges faster than (4) but the accelerated algorithm requires the knowledge of $\mu$ and $L$. Extrapolation techniques however obtain a similar convergence rate, but do not need estimates of the parameters $\mu$ and $L$. The idea is based on the comparison between the process followed by $\tilde{x}_i$ with a *linearized* model around the optimum (obtained by the first-order approximation of $\nabla f(x)$), written
$$x_{t+1} = x_t - \frac{1}{L}\Big( \underbrace{\nabla f(x^*)}_{=0} + \nabla^2 f(x^*)(x_t - x^*)\Big), \quad x_0 = \tilde{x}_0.$$

which can be rewritten as
$$x_{t+1} - x^* = (\mathbf{I} - \nabla^2 f(x^*)/L)(x_t - x^*), \quad x_0 = \tilde{x}_0. \tag{6}$$

A better estimate of the optimum in (6) can be obtained by forming a linear combination of the iterates (see [Anderson, 1965; Cabay and Jackson, 1976; Mešina, 1977]), with
$$\left\| \sum_{i=0}^{t} c_i x_i - x^* \right\| \ll \|x_t - x^*\|,$$

for some specific $c_i$ (either data driven, or derived from Chebyshev polynomials). These procedures were limited to quadratic functions only, i.e. when $\tilde{x}_i = x_i$ but this was recently extended to generic convex problems by Scieur et al. [2016] and we briefly recall these results below.

To simplify the notations, we write

$$\tilde{x}_{t+1} = g(\tilde{x}_t) \tag{7}$$

to be one step of algorithm $g$. We have that $g$ is differentiable, Lipchitz-continuous with constant $(1 - \kappa) < 1$, $g(x^*) = x^*$ and $g'(x^*)$ is symmetric. For example, the gradient method (3) matches exactly this definition with $g(x) = x - \nabla f(x)/L$. Running $k$ steps of (7) produces a sequence $\{\tilde{x}_0, ..., \tilde{x}_k\}$, which we extrapolate using Algorithm 1 from Scieur et al. [2016].

---

**Algorithm 1** Regularized Nonlinear Acceleration (**RNA**)

---

**Input:** Iterates $\tilde{x}_0, \tilde{x}_1, ..., \tilde{x}_{k+1} \in \mathbb{R}^d$ produced by (7), and a regularization parameter $\lambda > 0$.
  1: Compute $\tilde{R} = [\tilde{r}_0, ..., \tilde{r}_k]$, where $\tilde{r}_i = \tilde{x}_{i+1} - \tilde{x}_i$ is the $i^{th}$ residue.
  2: Solve

$$\tilde{c}^\lambda = \underset{c^T 1 = 1}{\operatorname{argmin}} \|\tilde{R}c\|^2 + \lambda\|c\|^2,$$

  or equivalently solve $(\tilde{R}^T\tilde{R} + \lambda\mathbf{I})z = \mathbf{1}$ then set $\tilde{c}^\lambda = z/\mathbf{1}^T z$.
**Output:** Approximation of $x^*$ computed as $\sum_{i=0}^{k} \tilde{c}_i^\lambda \tilde{x}_i$

---

For a good choice of $\lambda$, the output of Algorithm (1) is a much better estimate of the optimum than $\tilde{x}_{k+1}$ (or any other points of the sequence). Using a simple grid search on a few values of $\lambda$ is usually sufficient to improve convergence (see [Scieur et al., 2016] for more details).

## 3 Convergence of Regularized Nonlinear Acceleration

We quickly summarize the argument behind the convergence of Algorithm (1). The theoretical bound compares $\tilde{x}_i$ to the iterates produced by the linearized model

$$x_{t+1} = x^* + \nabla g(x^*)(x_t - x^*), \quad x_0 = \tilde{x}_0. \tag{8}$$

This sequence is useful to extend the convergence results to the nonlinear case, using sensitivity analysis. We write $c^\lambda$ the coefficients computed by Algorithm (1) from the "linearized" sequence $\{x_0, ..., x_{k+1}\}$ and the error term can be decomposed into three parts,

$$\left\|\sum_{i=0}^{k} \tilde{c}_i^\lambda \tilde{x}_i - x^*\right\| \leq \underbrace{\left\|\sum_{i=0}^{k} c_i^\lambda x_i - x^*\right\|}_{\text{Acceleration}} + \underbrace{\left\|\sum_{i=0}^{k}\left(\tilde{c}_i^\lambda - c_i^\lambda\right)(x_i - x^*)\right\|}_{\text{Stability}} + \underbrace{\left\|\sum_{i=0}^{k} \tilde{c}_i^\lambda\left(\tilde{x}_i - x_i\right)\right\|}_{\text{Nonlinearity}}. \tag{9}$$

Scieur et al. [2016] show that convergence is guaranteed as long as the errors $(\tilde{x}_i - x^*)$ and $(x_i - \tilde{x}_i)$ converge to zero fast enough, which ensures a good rate of decay for the regularization parameter $\lambda$, leading to an asymptotic rate equivalent to the accelerated rate in (5). In this section, we will use results from Scieur et al. [2016] to bound each individual term, but in this paper we improve the final convergence result.

The *stability* term (in $\tilde{c}^\lambda - c^\lambda$) is bounded using the *perturbation matrix*

$$P \triangleq R^T R - \tilde{R}^T \tilde{R}, \tag{10}$$

where $R$ and $\tilde{R}$ are the matrices of residuals,

$$R \triangleq [r_0...r_k] \qquad r_t = x_{t+1} - x_t, \tag{11}$$

$$\tilde{R} \triangleq [\tilde{r}_0...\tilde{r}_k] \qquad \tilde{r}_t = \tilde{x}_{t+1} - \tilde{x}_t. \tag{12}$$

The proofs of the following propositions were obtained by Scieur et al. [2016].

**Proposition 3.1** (Stability). *Let $\Delta c^\lambda = \tilde{c}^\lambda - c^\lambda$ be the gap between the coefficients computed by Algorithm (1) using the sequences $\{\tilde{x}_i\}$ and $\{x_i\}$ with regularization parameter $\lambda$. Let $P = R^T R - \tilde{R}^T \tilde{R}$ be defined in (10), (11) and (12). Then*

$$\|\Delta c^\lambda\| \leq \frac{\|P\|}{\lambda}\|c^\lambda\|. \tag{13}$$

*This implies that the stability term is bounded by*

$$\left\| \textstyle\sum_{i=0}^k \Delta c_i^\lambda (x_i - x^*)\right\| \leq \frac{\|P\|}{\lambda}\|c^\lambda\|\, O(\|x_0 - x^*\|). \tag{14}$$

The term **Nonlinearity** is bounded by the norm of the coefficients $\tilde{c}_\lambda$ (controlled thanks to the regularization parameter) times the norm of the *noise matrix*

$$\mathcal{E} = [x_0 - \tilde{x}_0,\ x_1 - \tilde{x}_1,\ ...,\ x_k - \tilde{x}_k]. \tag{15}$$

**Proposition 3.2** (Nonlinearity). *Let $\tilde{c}^\lambda$ be computed by Algorithm 1 using the sequence $\{\tilde{x}_0, ..., \tilde{x}_{k+1}\}$ with regularization parameter $\lambda$ and $\tilde{R}$ be defined in (12). The norm of $\tilde{c}^\lambda$ is bounded by*

$$\|\tilde{c}^\lambda\| \leq \sqrt{\frac{\|\tilde{R}\|^2 + \lambda}{(k+1)\lambda}} \leq \frac{1}{\sqrt{k+1}}\sqrt{1 + \frac{\|\tilde{R}\|^2}{\lambda}}. \tag{16}$$

*This bounds the nonlinearity term because*

$$\left\| \textstyle\sum_{i=0}^k \tilde{c}_i^\lambda (\tilde{x}_i - x_i)\right\| \leq \sqrt{1 + \frac{\|\tilde{R}\|^2}{\lambda}}\,\frac{\|\mathcal{E}\|}{\sqrt{k+1}}, \tag{17}$$

*where $\mathcal{E}$ is defined in (15).*

These two propositions show that the regularization in Algorithm 1 limits the impact of the noise: the higher $\lambda$ is, the smaller these terms are. It remains to control the *acceleration* term. For small $\lambda$, this term decreases as fast as the accelerated rate (5), as shown in the following proposition.

**Proposition 3.3** (Acceleration). *Let $\mathcal{P}_k$ be the subspace of real polynomials of degree at most $k$ and $S_\kappa(k, \alpha)$ be the solution of the Regularized Chebychev Polynomial problem,*

$$S_\kappa(k, \alpha) \triangleq \min_{p \in \mathcal{P}_k}\ \max_{x \in [0, 1-\kappa]}\ p^2(x) + \alpha\|p\|^2 \qquad s.t. \quad p(1) = 1. \tag{18}$$

*Let $\bar{\lambda} \triangleq \frac{\lambda}{\|x_0 - x^*\|^2}$ be the normalized value of $\lambda$. The acceleration term is bounded by*

$$\left\| \textstyle\sum_{i=0}^k c_i^\lambda x_i - x^* \right\| \leq \frac{1}{\kappa}\sqrt{S_\kappa(k, \bar{\lambda})\|x_0 - x^*\|^2 - \lambda\|c^\lambda\|^2}. \tag{19}$$

We also get the following corollary, which will be useful for the asymptotic analysis of the rate of convergence of Algorithm 1.

**Corollary 3.4.** *If $\lambda \to 0$, the bound (19) becomes*

$$\left\| \textstyle\sum_{i=0}^k c_i^\lambda x_i - x^* \right\| \leq \frac{1}{\kappa}\left(\frac{1 - \sqrt{\kappa}}{1 + \sqrt{\kappa}}\right)^k \|x_0 - x^*\|.$$

*Proof.* When $\lambda = 0$, (19) becomes $\frac{1}{\kappa}\sqrt{S_\kappa(k, 0)}\|x_0 - x^*\|$. The exact value of $\sqrt{S_\kappa(k, 0)}$ is obtained by using the coefficients of a re-scaled Chebyshev polynomial, derived by Golub and Varga [1961]; Scieur et al. [2016], and is equal to $\frac{1 - \sqrt{\kappa}}{1 + \sqrt{\kappa}}$. ∎

These last results controlling stability, nonlinearity and acceleration are proved by Scieur et al. [2016]. We now refine the final step of Scieur et al. [2016] to produce a global bound on the error that will allow us to extend these results to the stochastic setting in the next sections.

**Theorem 3.5.** *If Algorithm 1 is applied to the sequence $\tilde{x}_i$ with regularization parameter $\lambda$, it converges with rate*

$$\left\| \sum_{i=0}^k \tilde{c}_i^\lambda \tilde{x}_i \right\| \leq \|x_0 - x^*\| S_\kappa^{\frac{1}{2}}(k, \bar{\lambda})\sqrt{\frac{1}{\kappa^2} + \frac{O(\|x - x^*\|^2)\|P\|^2}{\lambda^3}} + \frac{\|\mathcal{E}\|}{\sqrt{k+1}}\sqrt{1 + \frac{\|\tilde{R}\|^2}{\lambda}}. \tag{20}$$

*Proof.* The proof is inspired by Scieur et al. [2016] and is straightforward. We can bound (9) using (14) (Stability), (17) (Nonlinearity) and (19) (Acceleration). It remains to maximize over the value of $\|c^\lambda\|$ using the result of Proposition A.2. ∎

This last bound is not very explicit, in particular because of the regularized Chebyshev term $S_\kappa(k, \bar\lambda)$. The solution is well known when $\bar\lambda = 0$ since it corresponds exactly to the rescaled Chebyshev polynomial [Golub and Varga, 1961], but as far as we know there is no known result about its regularized version, thus making the "finite-step" version hard to analyze. However, an asymptotic analysis simplifies it considerably. The next new proposition shows that when $x_0$ is close to $x^*$, then extrapolation converges as fast as in (5) in some cases.

**Proposition 3.6.** *Assume* $\|\tilde{R}\| = O(\|x_0 - x^*\|)$, $\|\mathcal{E}\| = O(\|x_0 - x^*\|^2)$ *and* $\|P\| = O(\|x_0 - x^*\|^3)$. *If we chose* $\lambda = O(\|x_0 - x^*\|^s)$ *with* $s \in [2, \frac{8}{3}]$ *then the bound* (20) *becomes*

$$\lim_{\|x_0 - x^*\| \to 0} \frac{\|\sum_{i=0}^k \tilde{c}_i^\lambda \tilde{x}_i\|}{\|x_0 - x^*\|} \le \frac{1}{\kappa}\left(\frac{1 - \sqrt{\kappa}}{1 + \sqrt{\kappa}}\right)^k.$$

*Proof. (Sketch)* The proof is based on the fact that $\lambda$ decreases slowly enough to ensure that the *Stability* and *Nonlinearity* terms vanish over time, but fast enough to have $\bar\lambda = \frac{\lambda}{\|x_0 - x^*\|^2} \to 0$. Then it remains to bound $S_\kappa(k, 0)$ with Corollary 3.4. The complete proof can be found in the Supplementary materials. ∎

**Note:** The assumptions are satisfied if we apply the gradient method on a twice differentiable, smooth and strongly convex function with Lipchitz-continuous Hessian [Scieur et al., 2016].

The efficiency of Algorithm 1 is thus ensured by two conditions. First, we need to be able to bound $\|\tilde{R}\|$, $\|P\|$ and $\|\mathcal{E}\|$ by decreasing quantities. Second, we have to find a proper rate of decay for $\lambda$ and $\bar\lambda$ such that the *stability* and *nonlinearity* terms go to zero when perturbations also go to zero. If these two conditions are met, then the accelerated rate in Proposition 3.6 holds.

## 4  Nonlinear and Noisy Updates

In (7) we defined $g(x)$ to be non linear, which generates a sequence $\tilde{x}_i$. We now consider noisy iterates

$$\tilde{x}_{t+1} = g(\tilde{x}_t) + \eta_{t+1}, \tag{21}$$

where $\eta_t$ is a stochastic noise. To simplify notations, we write (21) as

$$\tilde{x}_{t+1} = x^* + G(\tilde{x}_t - x^*) + \varepsilon_{t+1}, \tag{22}$$

where $\varepsilon_t$ is a stochastic noise (potentially correlated with the iterates $x_i$) with bounded mean $\nu_t$, $\|\nu_t\| \le \nu$ and bounded covariance $\Sigma_t \preceq (\sigma^2/d)\mathbf{I}$. We also assume $0\mathbf{I} \preceq G \preceq (1 - \kappa)\mathbf{I}$ and $G$ is symmetric. For example, (22) can be linked to (21) if we set $\varepsilon_t = \eta_t + O(\|\tilde{x}_t - x^*\|^2)$, which corresponds to the combination of the noise $\eta_{t+1}$ with the Taylor remainder of $g(x)$ around $x^*$, i.e.,

$$\tilde{x}_{t+1} = g(\tilde{x}_t) + \eta_{t+1} = \underbrace{g(x^*)}_{=x^*} + \underbrace{\nabla g(x^*)}_{=G}(\tilde{x}_t - x^*) + \underbrace{O(\|\tilde{x}_t - x^*\|) + \eta_{t+1}}_{=\epsilon_{t+1}}.$$

The recursion (22) is also valid when we apply the stochastic gradient method with fixed step size $h$ to the quadratic problem

$$\min_x \tfrac{1}{2}\|Ax - b\|^2.$$

This corresponds to (22) with $G = \mathbf{I} - hA^TA$ and mean $\nu = 0$. For the theoretical results, we will compare $\tilde{x}_t$ with their noiseless counterpart to control convergence,

$$x_{t+1} = x^* + G(x_t - x^*), \quad x_0 = \tilde{x}_0. \tag{23}$$

# 5 Convergence Analysis when Accelerating Stochastic Algorithms

We will control convergence in expectation. Bound (9) now becomes

$$\mathbb{E}\left[\left\|\sum_{i=0}^{k} \tilde{c}_i^\lambda \tilde{x}_i - x^*\right\|\right] \leq \left\|\sum_{i=0}^{k} c_i^\lambda x_i - x^*\right\| + O(\|x_0 - x^*\|)\mathbb{E}\left[\|\Delta c^\lambda\|\right] + \mathbb{E}\left[\|\tilde{c}^\lambda\|\|\mathcal{E}\|\right]. \quad (24)$$

We now need to enforce bounds (14), (17) and (19) in expectation. The proofs of the two next propositions are in the supplementary material. For simplicity, we will omit all constants in what follows.

**Proposition 5.1.** *Consider the sequences $x_i$ and $\tilde{x}_i$ generated by (21) and (23). Then,*

$$\begin{aligned}
\mathbb{E}[\|\tilde{R}\|] &\leq O(\|x_0 - x^*\|) + O(\nu + \sigma), & (25) \\
\mathbb{E}[\|\mathcal{E}\|] &\leq O(\nu + \sigma), & (26) \\
\mathbb{E}[\|P\|] &\leq O((\sigma + \nu)\|x_0 - x^*\|) + O((\nu + \sigma)^2). & (27)
\end{aligned}$$

We define the following *stochastic condition number*

$$\tau \triangleq \frac{\nu + \sigma}{\|x_0 - x^*\|}.$$

The Proposition 5.2 gives the result when injecting these bounds in (24).

**Proposition 5.2.** *The accuracy of extrapolation Algorithm 1 applied to the sequence $\{\tilde{x}_0, ..., \tilde{x}_k\}$ generated by (21) is bounded by*

$$\frac{\mathbb{E}\left[\|\sum_{i=0}^{k} \tilde{c}_i^\lambda \tilde{x}_i - x^*\|\right]}{\|x_0 - x^*\|} \leq \left(S_\kappa(k, \bar{\lambda})\sqrt{\frac{1}{\kappa^2} + \frac{O(\tau^2(1+\tau)^2)}{\bar{\lambda}^3}} + O\left(\sqrt{\tau^2 + \frac{\tau^2(1+\tau^2)}{\bar{\lambda}}}\right)\right). \quad (28)$$

Consider a situation where $\tau$ is small, e.g. when using stochastic gradient descent with fixed step-size, with $x_0$ far from $x^*$. The following proposition details the dependence between $\bar{\lambda}$ and $\tau$ ensuring the upper convergence bound remains stable when $\tau$ goes to zero.

**Proposition 5.3.** *When $\tau \to 0$, if $\bar{\lambda} = \Theta(\tau^s)$ with $s \in ]0, \frac{2}{3}[$, we have the accelerated rate*

$$\mathbb{E}\left[\|\sum_{i=0}^{k} \tilde{c}_i^\lambda \tilde{x}_i - x^*\|\right] \leq \frac{1}{\kappa}\left(\frac{1 - \sqrt{\kappa}}{1 + \sqrt{\kappa}}\right)^k \|x_0 - x^*\|. \quad (29)$$

*Moreover, if $\lambda \to \infty$, we recover the averaged gradient,*

$$\mathbb{E}\left[\|\sum_{i=0}^{k} \tilde{c}_i^\lambda \tilde{x}_i - x^*\|\right] = \mathbb{E}\left[\left\|\frac{1}{k+1}\sum_{i=0}^{k} \tilde{x}_i - x^*\right\|\right].$$

*Proof.* Let $\bar{\lambda} = \Theta(\tau^s)$, using (28) we have

$$\begin{aligned}
\mathbb{E}\left[\left\|\sum_{i=0}^{k} \tilde{c}_i^\lambda \tilde{x}_i - x^*\right\|\right] &\leq \|x_0 - x^*\|S_\kappa(k, \tau^s)\sqrt{\frac{1}{\kappa^2}O(\tau^{2-3s}(1+\tau)^2)} \\
&\quad + \|x_0 - x^*\|O(\sqrt{\tau^2 + \tau^{2-3s}(1+\tau^2)}).
\end{aligned}$$

Because $s \in ]0, \frac{2}{3}[$, means $2 - 3s > 0$, thus $\lim_{\tau \to 0} \tau^{2-3s} = 0$. The limits when $\tau \to 0$ is thus exactly (29). If $\lambda \to \infty$, we have also

$$\lim_{\lambda \to \infty} \tilde{c}_\lambda = \lim_{\lambda \to \infty} \mathrm{argmin}_{c:\mathbf{1}^T c = 1} \|\tilde{R}c\| + \lambda\|c\|^2 = \mathrm{argmin}_{c:\mathbf{1}^T c = 1} \|c\|^2 = \frac{1}{k+1}$$

which yields the desired result. ∎

Proposition 5.3 shows that Algorithm 1 is thus asymptotically optimal provided $\lambda$ is well chosen because it recovers the accelerated rate for smooth and strongly convex functions when the perturbations goes to zero. Moreover, we recover Proposition 3.6 when $\epsilon_t$ is the Taylor remainder, i.e. with $\nu = O(\|x_0 - x^*\|^2)$ and $\sigma = 0$, which matches the deterministic results.

Algorithm 1 is particularly efficient when combined with a restart scheme [Scieur et al., 2016]. From a theoretical point of view, the acceleration peak arises for small values of $k$. Empirically, the

improvement is usually more important at the beginning, i.e. when $k$ is small. Finally, the algorithmic complexity is $O(k^2 d)$, which is linear in the problem dimension when $k$ remains bounded.

The benefits of extrapolation are limited in a regime where the noise dominates. However, when $\tau$ is relatively small then we can expect a significant speedup. This condition is satisfied in many cases, for example at the initial phase of the stochastic gradient descent or when optimizing a sum of functions with variance reduction techniques, such as SAGA or SVRG.

## 6    Numerical Experiments

### 6.1    Stochastic gradient descent

We want to solve the least-squares problem

$$\min_{x \in \mathbb{R}^d} F(x) = \frac{1}{2} \|Ax - b\|^2,$$

where $A^T A$ satisfies $\mu \mathbf{I} \preceq (A^T A) \preceq L\mathbf{I}$. To solve this problem, we have access to the stochastic first-order oracle

$$\nabla_\varepsilon F(x) = \nabla F(x) + \varepsilon,$$

where $\varepsilon$ is a zero-mean noise of covariance matrix $\Sigma \preceq \frac{\sigma^2}{d}\mathbf{I}$. We will compare several methods.

- **SGD.** Fixed step-size, $x_{t+1} = x_t - \frac{1}{L}\nabla_\varepsilon F(x_t)$.
- **Averaged SGD.** Iterate $x_k$ is the mean of the $k$ first iterations of SGD.
- **AccSGD.** The optimal two-step algorithm in Flammarion and Bach [2015], with optimal parameters (this implies $\|x_0 - x^*\|$ and $\sigma$ are known exactly).
- **RNA+SGD.** The regularized nonlinear acceleration Algorithm 1 applied to a sequence of $k$ iterates of SGD, with $k = 10$ and $\lambda = \|\tilde{R}^T \tilde{R}\|/10^{-6}$.

By Proposition 5.2, we know that RNA+SGD will *not* converge to arbitrary precision because the noise is additive with a non-vanishing variance. However, Proposition 5.3 predicts an improvement of the convergence at the beginning of the process. We illustrate this behavior in Figure 1. We clearly see that at the beginning, the performance of RNA+SGD is comparable to that of the optimal accelerated algorithm. However, because of the restart strategy, in the regime where the level of noise becomes more important the acceleration becomes less effective and finally the convergence stalls, as for SGD. Of course, for practical purposes, the first regime is the most important because it effectively minimizes the generalization error [Défossez and Bach, 2015; Jain et al., 2016].

### 6.2    Finite sums of functions

We focus on the composite problem $\min_{x \in \mathbb{R}^d} F(x) = \sum_{i=1}^N \frac{1}{N} f_i(x) + \frac{\mu}{2}\|x\|^2$, where $f_i$ are convex and $L$-smooth functions and $\mu$ is the regularization parameter. We will use classical methods for minimizing $F(x)$ such as SGD (with fixed step size), SAGA [Defazio et al., 2014], SVRG [Johnson and Zhang, 2013], and also the accelerated algorithm Katyusha [Allen-Zhu, 2016]. We will compare their performance with and without the (potential) acceleration provided by Algorithm 1 with restart after $k$ data passes. The parameter $\lambda$ is found by a grid search of size $k$, the size of the input sequence, but it adds only *one* data pass at each extrapolation. Actually, the grid search can be faster if we approximate $F(x)$ with fewer samples, but we choose to present Algorithm 1 in its simplest version. We set $k = 10$ for all the experiments.

In order to balance the complexity of the extrapolation algorithm and the optimization method we wait several data queries before adding the current point (the "snapshot") of the method to the sequence. Indeed, the extrapolation algorithm has a complexity of $O(k^2 d) + O(N)$ (computing the coefficients $\tilde{c}^\lambda$ and the grid search over $\lambda$). If we wait at least $O(N)$ updates, then the extrapolation method is of the same order of complexity as the optimization algorithm.

- **SGD.** We add the current point after $N$ data queries (i.e. one epoch) and $k$ snapshots of SGD cost $kN$ data queries.

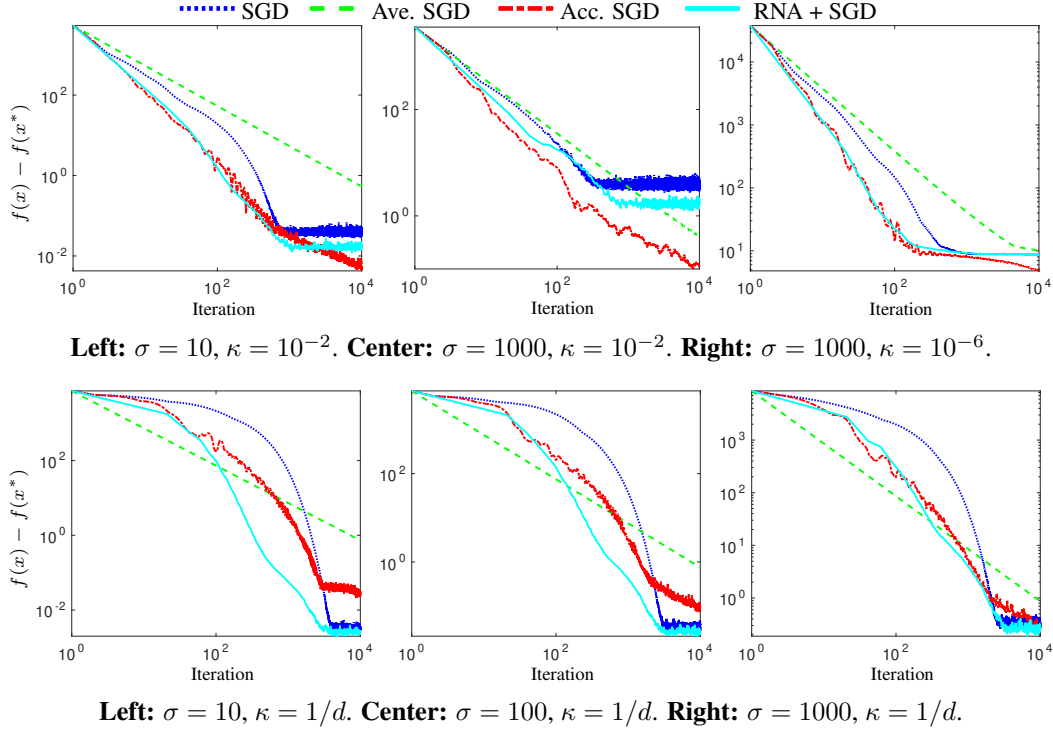

**Left:** $\sigma = 10$, $\kappa = 10^{-2}$. **Center:** $\sigma = 1000$, $\kappa = 10^{-2}$. **Right:** $\sigma = 1000$, $\kappa = 10^{-6}$.

**Left:** $\sigma = 10$, $\kappa = 1/d$. **Center:** $\sigma = 100$, $\kappa = 1/d$. **Right:** $\sigma = 1000$, $\kappa = 1/d$.

Figure 1: Comparison of performance between SGD, averaged SGD, Accelerated SGD [Flammarion and Bach, 2015] and RNA+SGD. We tested the performance on a matrix $A^T A$ of size $d = 500$, with (**top**) random eigenvalues between $\kappa$ and 1 and (**bottom**) decaying eigenvalues from 1 to $1/d$. We start at $\|x_0 - x^*\| = 10^4$, where $x_0$ and $x^*$ are generated randomly.

- **SAGA.** We compute the gradient table exactly, then we add a new point after $N$ queries, and $k$ snapshots of SAGA cost $(k + 1)N$ queries. Since we optimize a sum of quadratic or logistic losses, we used the version of SAGA which stores $O(N)$ scalars.

- **SVRG.** We compute the gradient exactly, then perform $N$ queries (the inner-loop of SVRG), and $k$ snapshots of SVRG cost $2kN$ queries.

- **Katyusha.** We compute the gradient exactly, then perform $4N$ gradient calls (the inner-loop of Katyusha), and $k$ snapshots of Katyusha cost $3kN$ queries.

We compare these various methods for solving least-squares regression and logistic regression on several datasets (Table 1), with several condition numbers $\kappa$: well ($\kappa = 100/N$), moderately ($\kappa = 1/N$) and badly ($\kappa = 1/100N$) conditioned. In this section, we present the numerical results on Sid (Sido0 dataset, where $N = 12678$ and $d = 4932$) with bad conditioning, see Figure 2. The other experiments are highlighted in the supplementary material.

In Figure 2, we clearly see that both SGD and RNA+SGD do not converge. This is mainly due to the fact that we do not average the points. In any case, except for quadratic problems, the averaged version of SGD does not converge to the minimum of $F$ with arbitrary precision.

We also notice that Algorithm 1 is unable to accelerate Katyusha. This issue was already raised by Scieur et al. [2016]: when the algorithm has a momentum term (like Nesterov's method), the underlying dynamical system is harder to extrapolate, in particular because the matrix presents in the linearized version of such systems is not symmetric.

Because the iterates of SAGA and SVRG have low variance, their accelerated version converges faster to the optimum, and their performance are then comparable to Katyusha. In our experiments, Katyusha was faster than RNA+SAGA only once, when solving a least square problem on Sido0

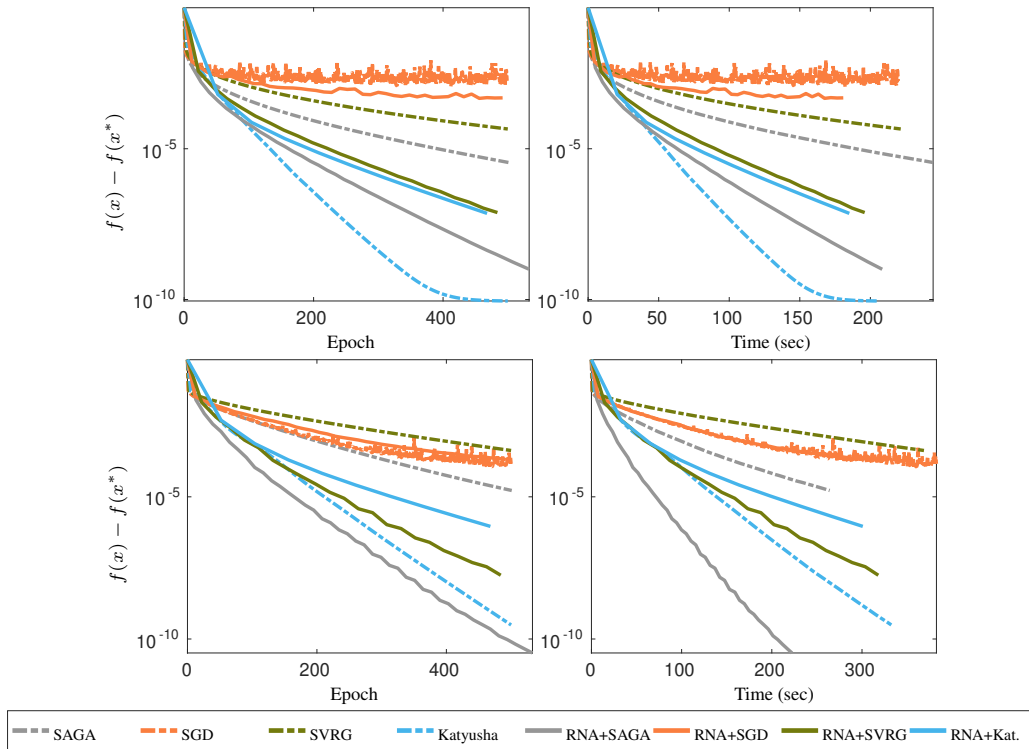

Figure 2: Optimization of quadratic loss (**Top**) and logistic loss (**Bottom**) with several algorithms, using the `Sid` dataset with bad conditioning. The experiments are done in Matlab. **Left:** Error vs epoch number. **Right:** Error vs time.

with a bad condition number. Recall however that the acceleration Algorithm 1 does not require the specification of the strong convexity parameter, unlike Katyusha.

## Acknowledgments

The authors would like to acknowledge support from a starting grant from the European Research Council (ERC project SIPA), from the European Union's Seventh Framework Programme (FP7-PEOPLE-2013-ITN) under grant agreement number 607290 SpaRTaN, as well as support from the chaire *Économie des nouvelles données* with the *data science* joint research initiative with the *fonds AXA pour la recherche* and a gift from Société Générale Cross Asset Quantitative Research.

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
