[Supplementary Material · Nonlinear_Acceleration_of_Stochastic_Algorithms_SuppMat.pdf]

# Supplementary material

## A    Missing propositions

**Proposition A.1.** *Let $\mathcal{E}$ be a matrix formed by $[\epsilon_1, \epsilon_2, ..., \epsilon_k]$, where $\epsilon_i$ has mean $\|\nu_i\| \leq \nu$ and variance $\Sigma_i \preceq \sigma \boldsymbol{I}$. By triangle inequality then Jensen's inequality, we have*

$$\mathbb{E}[\|\mathcal{E}\|_2] \leq \sum_{i=0}^{k} \mathbb{E}[\|\varepsilon_i\|] \leq \sum_{i=0}^{k} \sqrt{\mathbb{E}[\|\varepsilon_i\|^2]} \leq O(\nu + \sigma). \tag{30}$$

**Proposition A.2.** *Consider the function*

$$f(x) = \frac{1}{\kappa}\sqrt{a - \lambda x^2} + bx$$

*defined for $x \in [0, \sqrt{a/\lambda}]$. The its maximal value is attained at*

$$x_{opt} = \frac{b\sqrt{a}}{\sqrt{\frac{\lambda^2}{\kappa^2} + \lambda b^2}},$$

*and its maximal value is thus, if $x_{opt} \in [0, \sqrt{a/\lambda}]$,*

$$f_{\max} = \sqrt{a}\sqrt{\frac{1}{\kappa^2} + \frac{b^2}{\lambda}}. \tag{31}$$

*Proof.* The (positive) root of the derivative of $f$ follows

$$b\sqrt{a - \lambda x^2} - \frac{1}{\kappa}\lambda x = 0 \qquad \Leftrightarrow \qquad x = \frac{b\sqrt{a}}{\sqrt{\frac{\lambda^2}{\kappa^2} + \lambda b^2}}.$$

If we inject the solution in our function, we obtain its maximal value,

$$
\begin{aligned}
\frac{1}{\kappa}\sqrt{a - \lambda\left(\frac{b\sqrt{a}}{\sqrt{\frac{\lambda^2}{\kappa^2} + \lambda b^2}}\right)^2} + b\frac{b\sqrt{a}}{\sqrt{\frac{\lambda^2}{\kappa^2} + \lambda b^2}} &= \frac{1}{\kappa}\sqrt{a - \lambda\frac{b^2 a}{\frac{\lambda^2}{\kappa^2} + \lambda b^2}} + b\frac{b\sqrt{a}}{\sqrt{\frac{\lambda^2}{\kappa^2} + \lambda b^2}}, \\
&= \frac{1}{\kappa}\sqrt{a - \lambda\frac{b^2 a}{\frac{\lambda^2}{\kappa^2} + \lambda b^2}} + b\frac{b\sqrt{a}}{\sqrt{\frac{\lambda^2}{\kappa^2} + \lambda b^2}}, \\
&= \frac{1}{\kappa}\sqrt{\frac{a\lambda^2 \frac{1}{\kappa^2}}{\frac{\lambda^2}{\kappa^2} + \lambda b^2}} + b\frac{b\sqrt{a}}{\sqrt{\frac{\lambda^2}{\kappa^2} + \lambda b^2}}, \\
&= \sqrt{a}\frac{\frac{1}{\kappa^2}\lambda + b^2}{\sqrt{\frac{\lambda^2}{\kappa^2} + \lambda b^2}}, \\
&= \frac{\sqrt{a}}{\lambda}\sqrt{\frac{\lambda^2}{\kappa^2} + \lambda b^2}.
\end{aligned}
$$

The simplification with $\lambda$ in the last equality concludes the proof.  ∎

## B    Missing proofs

### B.1    Proof of Proposition 3.6.

Let $\lambda = \|x_0 - x^*\|^s$. In this case, (20) becomes

$$\frac{\left\|\sum_{i=0}^{k} \tilde{c}_i^\lambda \tilde{x}_i\right\|}{\|x_0 - x^*\|} \leq S_\kappa(k, \|x_0 - x^*\|^{s-2})\sqrt{\frac{1}{\kappa^2} + \frac{O(1)\|P\|^2}{\|x_0 - x^*\|^{3s-2}}} + \frac{\|\mathcal{E}\|}{\|x_0 - x^*\|\sqrt{k+1}}\sqrt{1 + \frac{\|\tilde{R}\|^2}{\|x_0 - x^*\|^s}}.$$

By assumption, $\|\tilde{R}\| = O(\|x_0 - x^*\|)$, $\|\mathcal{E}\| = O(\|x_0 - x^*\|^2)$ and $O(\|P\|) = O(\|x_0 - x^*\|^3)$. The previous bound becomes

$$\frac{\left\|\sum_{i=0}^{k} \tilde{c}_i^\lambda \tilde{x}_i\right\|}{\|x_0 - x^*\|} \leq S_\kappa(k, \|x_0-x^*\|^{s-2})\sqrt{\frac{1}{\kappa^2} + O(\|x_0 - x^*\|^{8-3s})} + O\left(\sqrt{\|x_0 - x^*\|^2 + \|x_0 - x^*\|^{4-s}}\right).$$

Because $\lambda \in ]2, \frac{8}{3}[$, all exponents of $\|x_0 - x^*\|$ are positive. By consequence,

$$\lim_{\|x_0 - x^*\| \to 0} \frac{\left\|\sum_{i=0}^{k} \tilde{c}_i^\lambda \tilde{x}_i\right\|}{\|x_0 - x^*\|} \leq \frac{1}{\kappa} S_\kappa(k, 0).$$

Finally, the desired result is obtained by using Corollary 3.4.

## B.2 Proof of Proposition 5.1

*Proof.* First, we have to form the matrices $\tilde{R}$, $\mathcal{E}$ and $P$. We begin with $\mathcal{E}$, defined in (15). Indeed,

$$\begin{aligned}
\mathcal{E}_i = x_i - \tilde{x}_i \quad &\Rightarrow \quad \mathcal{E}_0 = 0, \\
&\mathcal{E}_1 = \varepsilon_1, \\
&\mathcal{E}_2 = \varepsilon_2 + G\varepsilon_1, \\
&\mathcal{E}_k = \sum_{i=1}^{k} G^{k-i}\varepsilon_i.
\end{aligned}$$

It means that each $\|\mathcal{E}_i\| = O(\|\varepsilon_i\|)$. By using (30),

$$\begin{aligned}
\mathbb{E}\|\mathcal{E}\| &\leq \sum_i \mathbb{E}\|\mathcal{E}_i\| \\
&\leq \sum_i \mathbb{E}\|\mathcal{E}_i - \nu_i\| + \|\nu_i\| \\
&\leq O(\nu + \sigma)
\end{aligned}$$

For $\tilde{R}$, we notice that

$$\begin{aligned}
\tilde{R}_t &= \tilde{x}_{t+1} - \tilde{x}_t, \\
&= x_{t+1} - x_t + \mathcal{E}_{t+1} - \mathcal{E}_t, \\
&= R_t + \sum_{i=1}^{t+1} G^{t+1-i}\varepsilon_i - \sum_{i=1}^{t} G^{t-i}\varepsilon_i, \\
&= R_t + (G - \mathbf{I})\sum_{i=1}^{t} G^{t-i}\varepsilon_i + \varepsilon_{t+1} \\
&\leq O(\|x_0 - x^*\|) + O(\textstyle\sum_i \varepsilon_i).
\end{aligned}$$

We get (25) by splitting the norm,

$$\mathbb{E}[\|\tilde{R}\|] \leq \|R\| + \sum_{i=1}^{k} \mathbb{E}\left[O(\|\varepsilon_i\|)\right] \leq O(\|x_0 - x^*\|) + O(\nu + \sigma).$$

Finally, by definition of $P$,

$$\|P\| \leq 2\|\mathcal{E}\|\|R\| + \|\mathcal{E}\|^2.$$

Taking the expectation leads to the desired result,

$$\begin{aligned}
\mathbb{E}[\|P\|] &\leq 2\mathbb{E}[\|\mathcal{E}\|\|R\|] + \mathbb{E}[\|\mathcal{E}\|^2], \\
&\leq 2\|R\|\,\mathbb{E}[\|\mathcal{E}\|] + \mathbb{E}[\|\mathcal{E}\|_F^2], \\
&\leq O\left(\|x_0 - x^*\|(\sigma + \nu)\right) + O\left((\sigma + \nu)^2\right).
\end{aligned}$$

∎

## B.3 Proof of Proposition 5.2

*Proof.* We start with (24), then we use (13)

$$\|\sum_{i=0}^{k} c_i^\lambda x_i - x^*\| + O(\|x_0 - x^*\|)\mathbb{E}\Big[\|\Delta c^\lambda\|\Big] + \mathbb{E}\Big[\|\tilde{c}^\lambda\|\|\mathcal{E}\|\Big],$$

$$\leq \|\sum_{i=0}^{k} c_i^\lambda x_i - x^*\| + O(\|x_0 - x^*\|)\frac{\|c_\lambda\|}{\lambda}\mathbb{E}\Big[\|P\|\Big] + \sqrt{\mathbb{E}\Big[\|\tilde{c}_\lambda\|^2\Big]\mathbb{E}\Big[\|\mathcal{E}\|^2\Big]}.$$

The first term can be bounded by (19),

$$\|\sum_{i=0}^{k} c_i^\lambda x_i - x^*\| \leq \frac{1}{\kappa}\sqrt{S_\kappa(k,\bar{\lambda})\|x_0 - x^*\|^2 - \lambda\|c^\lambda\|^2}.$$

We combine this bound with the second term by maximizing over $\|c^\lambda\|$. The optimal value is given in (31),

$$\|\sum_{i=0}^{k} c_i^\lambda x_i - x^*\| + O(\|x_0 - x^*\|)\frac{\|c_\lambda\|}{\lambda}\mathbb{E}\Big[\|P\|\Big] \leq \|x_0 - x^*\|S_\kappa(k,\bar{\lambda})\sqrt{\frac{1}{\kappa^2} + \frac{O(\|x - x^*\|^2)\mathbb{E}[\|P\|]^2}{\lambda^3}},$$

where $\bar{\lambda} = \lambda/\|x_0 - x^*\|^2$. Since, by Proposition 5.1,

$$\mathbb{E}[\|P\|]^2 \leq O\left((\nu + \sigma)^2 (\|x_0 - x^*\| + \nu + \sigma)^2\right),$$

we have

$$\|\sum_{i=0}^{k} c_i^\lambda x_i - x^*\| + O(\|x_0 - x^*\|)\frac{\|c_\lambda\|}{\lambda}\mathbb{E}\Big[\|P\|\Big]$$

$$\leq \|x_0 - x^*\|S_\kappa(k,\bar{\lambda})\sqrt{\frac{1}{\kappa^2} + \frac{O(\|x - x^*\|^2(\nu + \sigma)^2)(\|x_0 - x^*\| + \nu + \sigma)^2}{\lambda^3}} \qquad (32)$$

The last term can be bounded using (16),

$$\sqrt{\mathbb{E}\Big[\|c_\lambda\|^2\Big]\mathbb{E}\Big[\|\mathcal{E}\|^2\Big]} \leq O\left(\sqrt{\sum_{i=0}^{k}\|\mathcal{E}\|_i^2}\sqrt{\mathbb{E}\Big[\|\tilde{c}_\lambda\|^2\Big]}\right)$$

$$\leq O\left((\nu + \sigma)\sqrt{\mathbb{E}\Big[\|\tilde{c}_\lambda\|^2\Big]}\right)$$

$$\leq O\left((\nu + \sigma)\sqrt{\mathbb{E}\Big[1 + \frac{\|\tilde{R}\|^2}{\lambda}\Big]}\right)$$

$$\leq O\left((\nu + \sigma)\sqrt{1 + \frac{\mathbb{E}\Big[\|\tilde{R}\|_F^2\Big]}{\lambda}}\right)$$

However,

$$\mathbb{E}\Big[\|\tilde{R}\|_F^2\Big] = \sum_{i=0}^{k}\mathbb{E}\Big[\|\tilde{r}_i\|^2\Big]$$

$$= \sum_{i=0}^{k}\|r_i\|^2 + \mathbb{E}\Big[r_i^T\mathcal{E}_i + \|\mathcal{E}_i\|^2\Big]$$

$$\leq O\left(\|x_0 - x^*\|^2 + (\nu + \sigma)\|x_0 - x^*\| + (\nu + \sigma)^2\right)$$

$$\leq O\left(\|x_0 - x^*\| + (\nu + \sigma))^2\right)$$

Finally,

$$\sqrt{\mathbb{E}\Big[\|c_\lambda\|^2\Big]\mathbb{E}\Big[\|\mathcal{E}\|^2\Big]} \leq O\left((\nu + \sigma)\sqrt{1 + \frac{(\|x_0 - x^*\| + (\nu + \sigma))^2}{\lambda}}\right) \qquad (33)$$

We get (28) by summing (32) and (33), then by replace all $\frac{\nu+\sigma}{\|x_0-x^*\|}$ by $\tau$ and $\frac{\lambda}{\|x_0-x^*\|^2}$ by $\bar{\lambda}$. ∎

# C   Additional numerical experiments

## C.1   Legend

- - - SAGA    - - - Sgd    - - - SVRG    - - - Katyusha    ━━ RNA+SAGA    ━━ RNA+Sgd    ━━ RNA+SVRG    ━━ RNA+Kat.

## C.2   datasets

|  | Sonar UCI (`Son`) | Madelon UCI (`Mad`) | Random (`Ran`) | Sido0 (`Sid`) |
|---|---|---|---|---|
| # samples $N$ | 208 | 2000 | 4000 | 12678 |
| Dimension $d$ | 60 | 500 | 1500 | 4932 |

Table 1: Datasets used in the experiments.

## C.3 Quadratic loss

### C.3.1 Sonar dataset

Figure 3: Quadratic loss with (top to bottom) good, moderate and bad conditioning using `Son` dataset.

## C.3.2   Madelon dataset

Figure 4: Quadratic loss with (top to bottom) good, moderate and bad conditioning using `Mad` dataset.

## C.3.3 Random dataset

Figure 5: Quadratic loss with (top to bottom) good, moderate and bad conditioning using `Ran` dataset.

## C.3.4 Sido0 dataset

Figure 6: Quadratic loss with (top to bottom) good, moderate and bad conditioning using `Sid` dataset.

## C.4 Logistic loss

### C.4.1 Sonar dataset

Figure 7: Logistic loss with (top to bottom) good, moderate and bad conditioning using `Son` dataset.

## C.4.2 Madelon dataset

Figure 8: Logistic loss with (top to bottom) good, moderate and bad conditioning using `Mad` dataset.

## C.4.3 Random dataset

Figure 9: Logistic loss with (top to bottom) good, moderate and bad conditioning using `Ran` dataset.

### C.4.4 Sido0 dataset

Figure 10: Logistic loss with (top to bottom) good, moderate and bad conditioning using `Sid` dataset.