[Reviews · NeurIPS 2017]

Reviewer 1



The paper extends recent work of Scieur et al [2016] on nonlinear acceleration via extrapolation of sequences from deterministic to stochastic optimization. The work by Scieur itself generalizes and extends results developed in the late 60s and 70s from quadratics to non-quadratics (whence the name “nonlinear”). Sequence extrapolation methods seem to have been “forgotten” or simply “not in use” by the ML and optimization community until recently, and have some interesting theoretical and practical properties. For instance, nonlinear regularized acceleration (NRA) is capable to accelerate the sequence of iterates formed by the gradient descent method and obtain the optimal accelerated rate. This is done via what essentially amounts to a “bootstrapping” extrapolation process. Stochastic methods are the state of the art for optimization/training methods in ML, which is enough motivation to study sequence extrapolation methods for stochastic sequences, such as those formed from the iterates of SGD, SVRG, SDCA, SAGA, QUARTZ and so on. Hence, the topic of the paper is both timely and relevant. Unfortunately, if theory is to be followed, each extrapolation steps is very expensive. When applied to the ERM problem, it is linear in the number of training points, and quadratic in k where k is the current iteration counter. Hence, the cost is extremely prohibitive, which means that the theoretical results are not useful. However, the authors show through experiments that practical benefits of their method are observed when a restarting strategy is used on consecutive sequences of iterates of small/constant length (experiments were done for k=10). The paper would have been much stronger if these issues were resolved – if acceleration can be provably established for the restarting technique they use in practical experiments. Main issues: 1) Sentence on line 25 does not make sense. How is 1 – kappa an accelerated rate? 2) Why is \sqrt{S_\kappa(k,0)} equal to 1/kappa * [(1-sqrt{\kappa}) / (1 + \sqrt{\kappa}) ] ^k? I do not see a derivation of this in the paper. In other words, how does Cor 3.4 follow from (19)? 3) Cor 3.4: a factor of 1/\kappa seems to be missing here. 4) It should be mentioned in Section 3 which of the results were established in Scieur et al. [2016] and which are new. This is is not clear. 5) Prop 3.3: are the polynomials supposed to have real coefficients? 6) Why should we be interested in the sequence {x_t} produced by the linearized model (8)? We are primarily interested in the sequence provided by the original model x_{t+1} = g(x_t). This should be better motivated. 7) Prop 3.6: Should the fraction defining the rate be (1-sqrt{\kappa}) / (1+ \sqrt{\kappa}) instead of (1 - \kappa ) / (1 + \kappa )? Small issues: 8) 27: method 9) 29 (and elsewhere): “performance” instead of “performances” 10) A comma is missing at the end of all displayed equations followed by “where”. 11) 57: an -> the 12) What does O(x_0-x^*) mean in (14)? Should this be O(\|x_0-x^*\|)? 13) 68: The sentence needs to be reformulated. Say clearly that g is defined through (7)... 14) Say Algorithm 1 and not Algorithm (1). 15) 78: compares 16) Sometimes you use boldface I and sometimes standard I for the identity matrix. 17) 131: nonlinear or non-linear 18) 140: corresponds 19) 175 and 194: write \mathbb{R}^d, as in the intro. 20) 199: after every k iterations? 21) 199-200: Is this what you do: You add one data pass after every 10 data samples? If so, that is too much! 22) Many places: “least-square” -> “least-squares” 23) 226: like the -> like === post rebuttal feedback === I am keeping my ratings.

Reviewer 2



Summary of the paper ==================== In principle, the extrapolation algorithm devised by Scieur et al. (2016) forms a scheme which allows one to speed up a given optimization algorithm by averaging iterates. The current paper extends this idea to the stochastic case by using the same the averaging technique only this time for noisy iterates. After recalling the results for the deterministic case (as shown in Scieur et al. (2016)), a theoretical upper bound is derived, based on theses results, by quantifying the amount of noise introduced into the optimization process. Finally, numerical experiments which demonstrate the speed-up offered by this technique for least-square and finite sum problems are provided. Evaluation ========== Extending this technique for stochastic algorithms is a crucial step for this technique as this family of algorithms dominants many modern applications. From practical point of view, this seems to provide a significantly acceleration to some (but not all) of the state-of-the-art algorithms. Moreover, the theoretical analysis provided in the paper shows that this stochastic generalization effectively subsumes the deterministic case. However, this paper does not provide a satisfying analysis of the theoretical convergence rate expected by applying the Algorithm 1 on existing stochastic algorithms (especially, for finite sum problems). It is therefore hard to predict in what settings, for which algorithms and to what extent should we expect acceleration. In my opinion, this forms a major weakness point of this paper. General Comments ================ - As mentioned above, I would suggest providing a more comprehensive discussion which allows more intuition regarding the benefits of this technique with respect to other existing algorithms. - Consider adding a more precise statement to address the theoretical convergence rate expected when applying this technique to a given optimization algorithm with a given convergence rate. Minor Comments ============== L13: f(x) -> f ? L13: Smooth and strong convexity are not defined\addressed at this point of the paper. L25: depends on L31: Many algorithms do exhibit convergence properties which are adaptive to the effective strong convexity of the problem: vanilla gradient descent, SAG (https://arxiv.org/abs/1309.2388, Page 8, 'an interesting consequence'). L52: f(x) -> f ? L55: Might be somewhat confusing to diverse from the conventional way of defining condition number. L61: The linear approximation may not be altogether trivial (also may worth to mention that the gradient vanishes..) L72: in the description of Algorithm 1, using boldface font to denote the vector of all ones is not aligned with vector notation used throughout the paper, and thus, is not defined a-priori. L81: can you elaborate more on why 'Convergence is guaranteed as long as the errors' implies to 'ensures a good rate of decay for the regularization parameter'. L103: lambda.. L118: I would split the technical statement from the concrete application fro SGD. L121: Maybe elaborate more on the technical difficulty you encounter to prove a 'finite-step' version without lim? L136: G is symmetric? L137: corresponds? L137: I'd suggest further clarifying the connection with Taylor reminder by making it more concrete. L140: corresponds L140: h in I-h A^\top A L149: The Figure 2 Legend: the 'acc' prefix is a bit confusing as 'acceleration' in the context of continuous optimization this term is used somewhat differently. L227: Can you give more intuition on why 'the algorithm has a momentum term' implies 'underlying dynamical system is harder to extrapolate.'?